# Class Conflict: Diffuse Competition between Mammalian and Reptilian Predators

Christopher R. Dickman *, Aaron C. Greenville, Glenda M. Wardle and Jenna P. Bytheway

Desert Ecology Research Group, School of Life and Environmental Sciences, The University of Sydney, Camperdown, NSW 2006, Australia; aaron.greenville@sydney.edu.au (A.C.G.); glenda.wardle@sydney.edu.au (G.M.W.); jenna.bytheway@sydney.edu.au (J.P.B.)
* Correspondence: chris.dickman@sydney.edu.au; Tel.: +61-2-9351-2318

**Abstract:** (1) Diffuse competition affects per capita rates of population increase among species that exploit similar resources, and thus can be an important structuring force in ecological communities. Diffuse competition has traditionally been studied within taxonomically similar groups, although distantly related intraguild species are likely also to compete to some degree. (2) We assessed diffuse competition between mammalian and reptilian predators at sites in central Australia over 24 years. Specifically, we investigated the effect of dasyurid marsupial abundance on the diet breadth of three groups of lizards (nocturnal dietary generalists, diurnal dietary generalists and dietary specialists). (3) Nocturnal generalist lizards had progressively narrower diets as dasyurid abundance increased. The diet breadth of diurnal generalist lizards was unaffected by overall dasyurid abundance, but was restricted by that of the largest dasyurid species (*Dasycercus blythi*). Ant- and termite-specialist lizards were unaffected by dasyurid abundance. (4) Diffuse competition, mediated by interference, between dasyurids and nocturnal generalist lizards appears to have strong effects on these lizards, and is the first such between-class interaction to be described. Diffuse interactions may be widespread in natural communities, and merit further investigation among other disparate taxon groups that occur in the same ecological guilds.

**Keywords:** diffuse competition; dietary restriction; dasyurid marsupial; desert lizard; scat analysis; intraguild predation; interference competition

---

## 1. Introduction

Diverse ecological communities of different species that exploit similar resources in similar ways are likely to experience some degree of competition [1]. If these communities comprise largely predators, competitive interactions are likely to be mediated primarily by antagonism or interference, which, in extreme forms, may be manifest as intraguild killing or predation [2–4]. In recent years, competition and predation in multi-species communities have received some theoretical attention [5,6], with both direct and indirect trophic interactions emerging as important structuring forces [7,8]. Despite the importance of these interactions in shaping community structure, competition and predation in multi-species communities have received relatively little empirical attention, and experiments have typically been limited to pairs of species [9]. Experiments on competition between predators have generally been restricted further to couplets involving related species, although interactions between taxonomically disparate species have begun to emerge [10–14].

The importance of interactions within diverse multi-species communities has long been recognised [15, 16], but an important first step in formalising their strength and effects was taken by MacArthur [17], using the concept of diffuse competition. Here, a species' per capita rate of population increase is predicted to decline with increasing numbers of competitors, such that a constellation of species will

more readily outcompete a focal species than will a single competitor [18,19]. Over evolutionary time scales, diffuse competition could be expected to promote specialisation by competitor species on a narrow range of resources to which they have priority of access [20,21]. Over ecological time scales, by contrast, short-term changes in the abundances of dominant species within competition communities should drive temporary shifts in the range of resources to which inferior competitors have access [22,23]. Commonly, inferior competitors show reductions in diet breadth and in the range of habitats that they use when co-occurring dominant species are abundant compared with when they are scarce [24–26].

Diffuse competition has traditionally been studied within taxonomically similar groups, for example among species of plants, reptiles, birds or mammals. Competition within many plant communities is considered to be primarily diffuse [27,28], as it is among some fungal communities [29,30]. Diffuse competition has also been highlighted as an important influence on the distribution, abundance and resource use of diverse assemblages of desert ants [31,32], lizards [33,34] and riparian breeding birds [35]. Among multi-species communities of carnivorous mammals, competitive interactions have been observed to drive shifts in vigilance behaviour, movements and diet of the inferior—usually smaller—competitors [26,36–38]. In general, both direct and indirect interactions are more likely to occur in systems with many competing species [39]; the increased complexity in such systems also increases the practical difficulty of disentangling all the interactions.

Diffuse competition has rarely been investigated between distantly related intraguild species, despite the potential for it to occur. Here, we investigate the prevalence of diffuse competition within a guild comprising taxonomically disparate dasyurid marsupial (class Mammalia) and saurian (class Reptilia) predators in the Simpson Desert of central Australia. Predators are often diverse and abundant in arid environments [40,41], and could be expected to face pronounced fluctuations in prey resources between dry periods when primary productivity is low and rainy periods when productivity is briefly elevated [42]. The central deserts of Australia contain highly diverse assemblages of insectivorous mammals and lizards [23,43] and are subject to extreme temporal fluctuations in productivity [44], making them an ideal arena in which to investigate diffuse competition.

Dasyurid marsupials in arid Australia are mostly generalist insectivores, although they usually avoid ants and termites, and larger species include some small lizards and small mammals in their diet [45–48]. All species are primarily nocturnal, with minimal activity occurring near dawn and dusk [49]. Behavioural interference and competitive interactions have been suggested to drive niche separation between planigales (*Planigale gilesi* and *P. tenuirostris*) and fat-tailed dunnarts (*Sminthopsis crassicaudata*) [50], and the brush-tailed mulgara (*Dasycercus blythi*) appears to dominate smaller dasyurids when it is present in local communities [51]. Desert lizards are primarily insectivorous and consume a wide range of invertebrate taxa, but some species are considered ant- or termite-specialists [52,53]. They have diverse activity patterns, with some species being nocturnal while others are diurnal [54]. Diffuse competition may drive niche partitioning within assemblages of Australian desert lizards [33], with rare lizard species suggested to be uncommon due to diffuse competition from other more abundant species with high resource overlap [34]. No studies have yet explored interactions between desert lizards and dasyurids, despite the possibility of diffuse competition being an organising force within this extraordinarily rich and taxonomically disparate guild of small predators. Members of this guild interact with still richer assemblages of invertebrate predators such as spiders and scorpions (which they often hunt) and larger canid, felid and avian predators (to which lizards and dasyurids fall prey), although interactions between these trophic levels more likely represent predation [40,41,49] than the diffuse interactions that we focus on here.

To explore diffuse competition, we investigated the effect of dasyurid marsupial abundance on lizard diet breadth, focusing in particular on the diets of skinks and geckoes, the two most species-rich groups in Australia's central deserts. While we expected competition to occur, at least potentially, between all species given that they are all primarily insectivorous, we expected it to be strongest between those that are active at similar times in the diel cycle and thus are honing in on the same prey base. We assumed further that dasyurids are dominant over lizards given their larger size

(6–150 g vs. 0.05–30 g, respectively [55,56]) and higher metabolic rate demands [45,57,58]. Based on current knowledge of dasyurid and lizard diets and activity times, we predicted that:

1.  Nocturnal lizards with generalist diets will experience the strongest competition with dasyurids;
2.  Diurnal lizards with generalist diets will experience weak competition with dasyurids;
3.  Ant- and termite-specialist lizards will experience weak or no competition with dasyurids; and
4.  Competition, if it occurs, will be via interference and be detectable as intraguild predation.

If lizards experience competition from dasyurids, we expected this to be manifested in the lizards having narrower diets at times when dasyurids were abundant than when they were rare. We chose to use this aspect of resource use to detect competition because dietary shifts are likely to be especially sensitive to changes in the intensity of competition [23,59,60] and, unlike in traditional models of competition [61], food abundance (or carrying capacity, K) does not need to be known. In addition, the very low recapture rates of desert lizards preclude the estimation of demographic parameters [62] and hence reduce the utility of techniques that depend on reliable census data—such as regressing species' densities against each other—to derive competition or general interaction coefficients [63,64].

## 2. Materials and Methods

### 2.1. Study Site

The study area spanned approximately 8000 km$^2$ in the northeastern Simpson Desert, Queensland, Australia. Most fieldwork was carried out on Ethabuka Reserve (23°46′ S, 138°28′ E), with additional sampling on Carlo station and Cravens Peak Reserve immediately to the north, and Tobermorey station immediately to the west. The landscape is characterised by 8–10 m high, long parallel sand dunes that run in a NNW–SSE direction and lie 0.6–1 km apart [65]. Vegetation is dominated by spinifex grass (*Triodia basedowii*), which provides ground cover in the swales and on the dune sides. Gidgee trees (*Acacia georginae*) occur in low-lying clay soils in dune valleys, and shrubs such as *Acacia ligulata*, *Eucalyptus pachyphylla*, *Dodonaea viscosa*, *Grevillea stenobotrya* and *Grevillea juncifolia* occur sporadically throughout the study area [66,67]. The regional climate is highly irregular and driven by the El Niño Southern Oscillation [68]. During summer, daily temperatures usually exceed 40 °C and minimum temperatures often drop below 5 °C in winter [65]. Most rainfall occurs over the austral summer between November and February, and the long-term rainfall average is 199 mm/year (*n* = 94 years, recorded at Marion Downs, 120 km from the study area).

### 2.2. Field Sampling

Dasyurids, skinks and geckoes were captured at nine primary sites and seven supplementary sites (>1 km apart) over a period of 24 years (March 1990–February 2014) in the study area. Each site consisted of a 1 ha, 6 × 6 trapping grid comprising 36 pitfall traps (16 cm in diameter, 60 cm deep) spaced 20 m apart and buried flush to the ground. Trap efficiency was increased by erecting a drift fence of aluminium flywire (30 cm high, 5 m long) over the top of each trap to intercept surface-active animals and guide them toward the trap opening [69]. To prevent animals from digging their way out of the pitfall traps once captured, the bottom of the traps were covered with flywire screening. Pitfall traps were capped with metal lids when not in use. To ensure the topographic range of the dune field was sampled, the top lines of traps were established along the dune crest and the bottom lines 100 m distant in the dune valley. Traps were opened for 3 to 6 consecutive nights every 2–4 months throughout the 24 years of study, checked in the early morning and usually again in the late afternoon, and all captured animals were identified to species, weighed, their sex determined (where possible) and given a unique identification marker by ear-clipping (dasyurids) or toe-clipping (lizards). All animals were released within 5 m of their point of capture after processing.

Faecal pellets were collected from live individuals (both lizards and dasyurids) that defecated during handling. On occasion, lizards were held during the day in calico bags prior to release to

increase the likelihood that faecal pellets would be produced. Faecal pellets were also collected from pitfall traps, but only when there was no uncertainty of the donor (e.g., when a single animal had been captured in a clean and newly-opened trap). All faecal samples were placed into individually-labelled vials and air dried. Stomach samples were collected opportunistically from any individuals found dead in the field, and were preserved in 80% ethanol.

All field methods were carried out over the 24 year duration of the study under Scientific Purpose Permits from the relevant Queensland government department (W4/002533/00SAA, WO/000738/00/SAA, WISP02994105, WISP15192414, WITK15192514, WISP07623410 and WITK07635510) and with approval from The University of Sydney Animal Ethics Committee (L04/5-96/2/2361, L04/1-98/3/2656, L04/4-2000/1/3/3130, L04/2-2001/3/3344, L04/4-2004/3/3896, L04/1-2007/3/4510, L04/4-2009/3/5020 and L04/4-2010/3/5297).

### 2.3. Diet Analysis

The contents of faecal pellets and stomachs were systematically searched for distinguishable prey items that were classified as either invertebrate, vertebrate or plant material under a dissecting microscope. Invertebrates were further identified with reference to Bytheway and Dickman [70] to the taxonomic level of order, with the following exceptions: ants were separated from other Hymenoptera to the family level Formicidae; termites were identified to the infraorder Isoptera; centipedes to the class Chilopoda; snails and slugs to the class Gastropoda; and larvae were placed in their own category. The minimum number of individuals of each prey type was recorded, as determined by the number of heads, mandibles, wings, legs or pedipalps. Vertebrate remains were identified using hair, feather, scale, bone or other remains such as teeth, beaks or digits [47].

Faecal pellet analysis is a standard and effective method to determine diet without compromising the well-being of the study animals [71,72], although it can be subject to biases such as differential digestibility of different prey items. We minimised the possibility for bias against soft prey items [71], which are likely to be destroyed by digestive processes, by carefully searching for body parts such as head capsules. Diet analysis based on such meticulous faecal pellet examinations are highly comparable to those based on stomach content analysis [72]. Thus, we combined our faecal pellet and stomach content data for all analyses.

### 2.4. Statistical Analysis

The abundance of dasyurids was assessed using live-trapping results obtained at the same time and location that lizard diet samples were collected. Dasyurid captures were standardised per 100 trap nights and averaged for each month of trapping, following Greenville et al. [73].

Preliminary inspection of the dietary and live-trapping results for lizards, as well as comparison with previously published work [54,74,75], allowed us to confidently assign lizards to one of the three pre-identified categories (nocturnal dietary generalists, diurnal dietary generalists, dietary specialists). To determine whether a sufficient number of faecal/stomach samples had been analysed to accurately describe the overall diet of species in each category, we plotted the cumulative diversity of prey items identified in each sample against sample size. Diversity was calculated using the Brillouin index [76]. This index is suitable in being sensitive to rare species in samples, which was important in allowing us to quantify diet breadth, and is appropriate also when the randomness of samples cannot be guaranteed [77]. Diversity is calculated according to the equation:

$$H = \frac{\ln N! - \sum \ln n_i!}{N} \tag{1}$$

where $H$ is diversity, $N$ is the total number of individual prey recorded, and $n_i$ is the number of individual prey items in the $i$th category [76]. The Brillouin index takes on positive values above zero and increases with increasing diversity; although unbounded, calculated values seldom exceed 5. Following these analyses, the nocturnal generalist skink *Eremiascincus phantasmus* was separated from the nocturnal dietary generalist category, which otherwise contained gecko species.

To test our first three predictions, we conducted quantile regressions [78] to investigate changes in lizard diet breadth with increasing dasyurid abundance at the mean (0.5th quantile) and upper bound (0.9th quantile) for each of the three lizard categories. In addition, we assessed the effects separately of the most abundant dasyurid, the lesser hairy-footed dunnart (*Sminthopsis youngsoni*, ~9 g) which accounted for ~60% of all dasyurid captures, and the largest dasyurid, the brush-tailed mulgara (*Dasycercus blythi*, ~100 g). Quantile regressions are appropriate for our dataset as they are robust to outliers and skewed data distributions [79]. Furthermore, they are valuable when all factors affecting an organism cannot be measured and accounted for [79,80]. For the quantile regression analyses, diet breadth was described initially as prey diversity using the Brillouin index and also, for comparison, using species richness. Quantile regression analyses were performed in JMP$^{®}$ Pro 9.0.0 using the SAS (v. 9.4) add-in (SAS Institute Inc., Cary, NC, USA, 2010).

Where significant results were found in the quantile regressions, non-metric multi-dimensional scaling (nMDS) and global one-way analyses of similarities (ANOSIM), based on a Bray-Curtis similarity matrix [81] were conducted. These analyses allowed comparisons of presence-absence data for each prey category to investigate differences in prey species composition at low (0–1.99), moderate (2–3.99) and high (>4) levels of dasyurid abundance (captures per 100 trap nights). These analyses are based on the rank order of similarities between samples and do not assume a normal distribution, equal variances or covariances [82]. If a significant result was obtained from the ANOSIM, the prey categories that contributed most to the dissimilarity were determined using similarity percentages (SIMPER) [81]. nMDS, ANOSIM and SIMPER analyses were conducted using PRIMER Version 6.1.16 (PRIMER-E, Plymouth, UK).

To test our fourth prediction, we simply tallied the numbers of faecal/stomach samples of dasyurids that contained the remains of skinks or geckoes and expressed these as percentages of the total numbers of samples analysed.

## 3. Results

### 3.1. Diet Analyses

We analysed a total of 542 faecal pellets and stomachs from lizards, which included 138 samples from seven species of nocturnal dietary generalists, 263 samples from 16 species of diurnal dietary generalists and 141 samples from five species of dietary specialists (for a list of species refer to Table S1). The cumulative diversity ($H_k$) of prey taxa in the diet of nocturnal generalists, diurnal generalists and specialists all reached an asymptote at a sample size well below the number of faecal pellets and stomachs analysed (Figure 1), indicating that sample sizes were sufficient to reliably characterise dietary breadth for each category of lizard. A total of 24 prey taxa (Table S2) were identified in the lizard faecal and stomach samples. As expected, the diet breadth of specialist lizards was markedly less than that of the generalists (Figure 1), and comprised >95% termites for all species in this category. By contrast, individual generalist species included up to 18 of the 24 prey taxa in their diets. In some species, such as Bynoe's gecko (*Heteronotia binoei*), the representation of the different prey taxa was even (all taxa comprised <10% of the diet by frequency). In others such as the three-lined knob-tail gecko (*Nephrurus levis*), prey taxa such as spiders, crickets/grasshoppers and ants dominated the diet (18–22% by frequency), whereas mantids, earwigs, flies and silverfish were present but scarce (<1%). The diets of diurnal generalist lizards were slightly more diverse than those of nocturnal generalists (Figure 1), largely due to the occasional inclusion of taxa such as mites or pseudo-scorpions in the diets of the diurnal generalists.

Eight species of dasyurids (Table S3) were captured 4813 times at the study site over the 24 year sampling period in a sampling effort totalling 187,272 trap nights (trap success = 2.57%). In total, 224 faecal pellet samples were collected and analysed from these species.

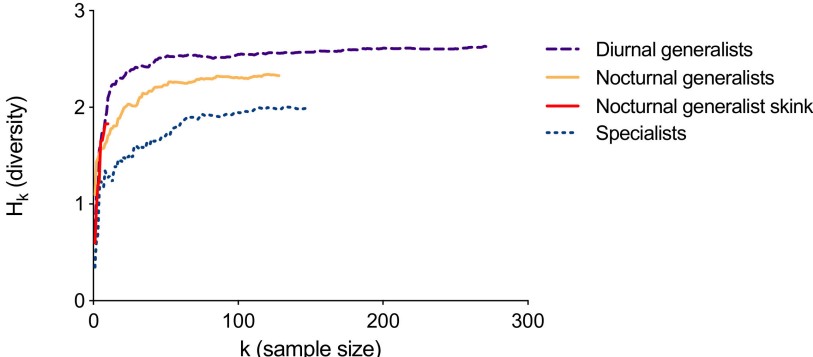

**Figure 1.** Cumulative diversity ($H_k$) of prey taxa in the diet of nocturnal generalists, nocturnal generalist skinks, diurnal generalists, and specialists with increasing sample size (k).

*3.2. Predictions 1–3*

Quantile regressions revealed a significant negative relationship between the diet breadth of nocturnal generalist lizards and dasyurid abundance at both the 0.5 and 0.9 quantiles (Table 1; Figure 2a). In contrast, the diet breadth of diurnal generalists and specialists was not affected by dasyurid abundance (Figure 2b,c, respectively). The relationship between lizard diet breadth and *S. youngsoni* (the most abundant dasyurid) yielded similar results for all hypothesis categories (Table 1). A significant negative relationship between the abundance of mulgara (*D. blythi*; the largest dasyurid) and the diet breadth of diurnal generalist lizards was revealed at the 0.5 quantile (Table 1; Figure 2d). Quantile regressions conducted using species richness to describe prey diversity yielded similar results, and indicated further that the diet breadth of nocturnal generalist lizards decreased from 5–6 taxa when dasyurids were scarce to just 1–2 taxa when dasyurids were abundant (Table S4 and Figure S1).

nMDS produced ordination plots with stress values <0.2, indicating that the plots (Figure S2) may be interpreted reliably [83]. While there was no obvious separation of samples in the plots, global one-way ANOSIMs revealed that the prey species composition of lizard diets differed between the three categories of dasyurid abundance (global R = 0.121, 0.137, and 0.189, *p* = 0.003, 0.001, and 0.031 for nocturnal generalists with total dasyurid abundance and *S. youngsoni* abundance, and diurnal generalists with mulgara abundance, respectively). SIMPER analyses revealed that spiders (17%; 17%), followed by crickets/grasshoppers (14%; 15%) and beetles (11%; 12%) were the greatest contributing factors to the dissimilarity in nocturnal generalist diets at low and high dasyurid abundance for both total dasyurid and *S. youngsoni* abundance, respectively. In both cases, as dasyurid abundance increased and nocturnal generalist diet breadth decreased, fewer of these prey taxa were consumed by the lizards. In contrast, SIMPER showed that more true bugs (17%), crickets/grasshoppers (11%) and ants (11%) were eaten by diurnal generalist lizards as mulgara abundance increased.

*3.3. Prediction 4*

Skink or gecko remains were found in 14 of 72 faecal pellets from *D. blythi* (19.4%), in 2 of 60 faecal pellets from *S. youngsoni* (3.3%) and in 1 of 27 faecal pellets from *S. hirtipes* (3.7%). Of the 17 lizard remains detected, 12 were identified as gecko; 11 of these were recovered from the faecal pellets of *D. blythi* and the last from a faecal pellet of *S. youngsoni*. Analysis of a further 65 pellets from the other five species of dasyurids in the study system failed to detect any lizard remains. By comparison, the remains of a juvenile rodent were found in just one of the 542 lizard diet samples, that of a nocturnal generalist, *Nephrurus levis*.

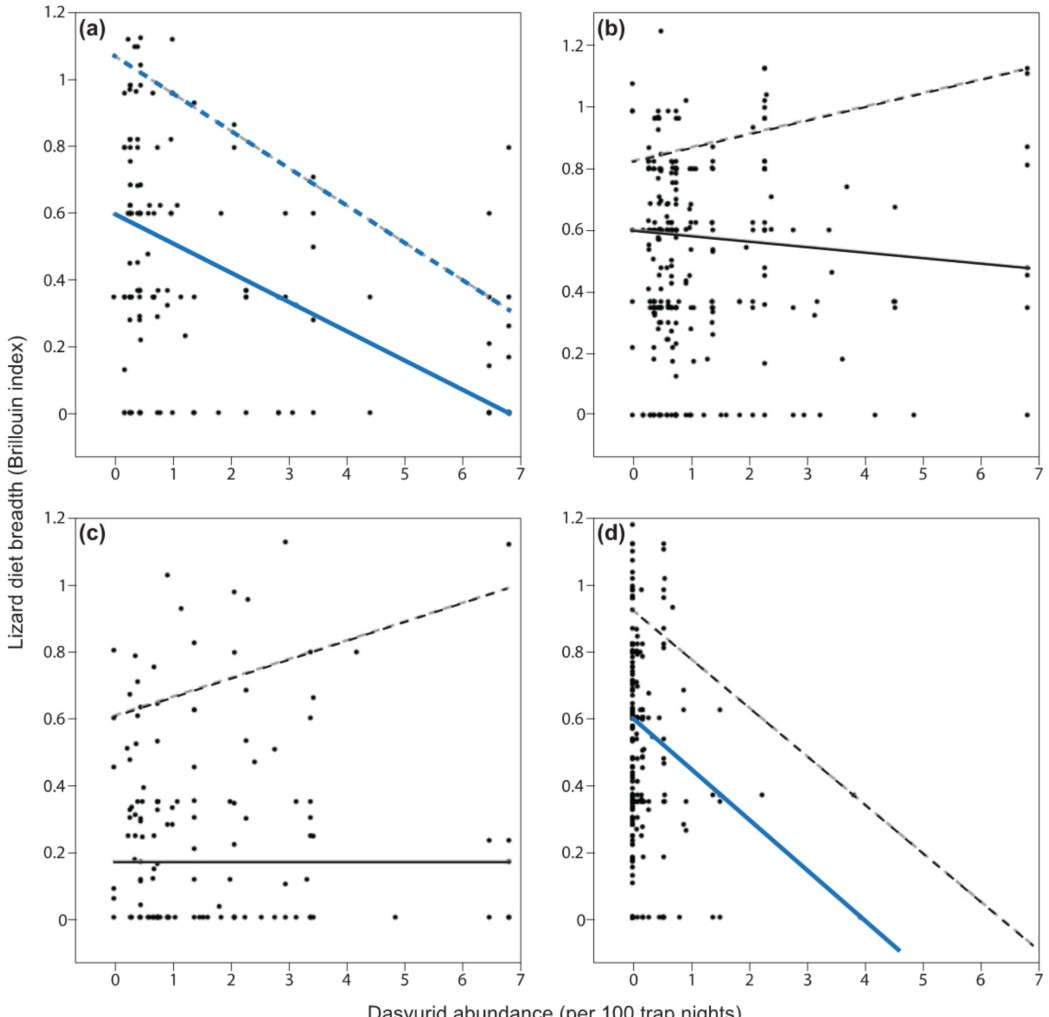

**Figure 2.** Relationships between lizard diet breadth and dasyurid abundance (all species) at the 0.5 (solid line) and 0.9 (dashed line) quantiles for (**a**) nocturnal generalist lizards, (**b**) diurnal generalist lizards and (**c**) specialist lizards, and (**d**) mulgara (*Dasycercus blythi*) abundance and diurnal generalist lizards. Significant relationships ($p < 0.001$) are shown in blue (in panel (a), both 0.5 and 0.9 quantiles are significant; and in panel (d), it is the 0.5 quantile only).

**Table 1.** Results of quantile regression models comparing the 0.5 and 0.9 quantiles of lizard diet breadth (Brillouin index) as a function of dasyurid abundance for a suite of predictions. Prediction categories for lizard groupings are based on activity period (nocturnal or diurnal) and diet breadth (generalist vs. specialist); and for mammalian competitors are grouped by all dasyurids or separately for each of two species, the most abundant (*Sminthopsis youngsoni*) and the largest (*Dasycercus blythi*) in the dasyurid community. A significant *p*-value indicated by * and bold type shows that there is a non-zero slope for that quantile.

| Prediction Category—Lizard Group Dasyurid Group | 0.5 Quantile | | | | 0.9 Quantile | | | |
|---|---|---|---|---|---|---|---|---|
| | Estimate | CI | *t*-Value | *p* | Estimate | CI | *t*-Value | *p* |
| **Nocturnal generalists** | | | | | | | | |
| All dasyurids | −0.087 | −0.102–0.072 | −11.478 | **<0.0001 *** | −0.112 | −0.150–0.073 | −5.739 | **<0.0001 *** |
| *Sminthopsis youngsoni* | −0.111 | −0.131–0.091 | −10.933 | **<0.0001 *** | −0.136 | −0.176–0.097 | −6.799 | **<0.0001 *** |
| *Dasycercus blythi* | −0.197 | −0.612–0.219 | −0.937 | 0.3506 | 0.157 | −0.232–0.545 | 0.799 | 0.4256 |

**Table 1.** *Cont.*

| Prediction Category—Lizard Group Dasyurid Group | 0.5 Quantile | | | | 0.9 Quantile | | | |
|---|---|---|---|---|---|---|---|---|
| | Estimate | CI | *t*-Value | *p* | Estimate | CI | *t*-Value | *p* |
| **Nocturnal generalist skink** | | | | | | | | |
| All dasyurids | 0.018 | −0.673–0.708 | 0.060 | 0.9539 | 0.171 | −0.978–1.320 | 0.343 | 0.7406 |
| *Sminthopsis youngsoni* | −0.959 | −4.413–2.496 | −0.640 | 0.5402 | −1.751 | −4.563–1.061 | −1.436 | 0.189 |
| *Dasycercus blythi* | 0.049 | −0.948–1.047 | 0.114 | 0.9121 | 0.235 | −2.530–3.000 | 0.196 | 0.8494 |
| **Diurnal generalists** | | | | | | | | |
| All dasyurids | −0.018 | −0.076–0.040 | −0.607 | 0.5441 | 0.044 | −0.002–0.090 | 1.880 | 0.0612 |
| *Sminthopsis youngsoni* | 0.039 | −0.031–0.109 | 1.103 | 0.2712 | 0.057 | −0.001–0.116 | 1.932 | 0.0545 |
| *Dasycercus blythi* | −0.152 | −0.232–0.072 | −3.746 | **0.0002** * | −0.145 | −0.516–0.226 | −0.771 | 0.4416 |
| **Specialists** | | | | | | | | |
| All dasyurids | <0.001 | −0.044–0.044 | <0.001 | 1.0000 | 0.056 | −0.028–0.140 | 1.321 | 0.1886 |
| *Sminthopsis youngsoni* | 0.000 | −0.058–0.058 | 0.000 | 1.0000 | 0.094 | −0.033–0.222 | 1.467 | 0.1445 |
| *Dasycercus blythi* | −0.042 | −0.236–0.151 | −0.433 | 0.6654 | 0.181 | −0.256–0.618 | 0.819 | 0.4143 |

## 4. Discussion

Our results suggest that diffuse competition occurs between desert lizards and dasyurids, and also that competition is stronger between some components of this taxonomically disparate guild of small predators than others. We found that nocturnal generalist lizards experienced a marked narrowing of diet breadth as dasyurid abundance increased, suggesting strong competition between these members of the guild and providing support for our first prediction. Overall, dasyurid abundance did not affect the diet breadth of diurnal generalist lizards, but their diet was restricted by the largest dasyurid species (brush-tailed mulgara, *D. blythi*), thus providing only partial support for our second prediction. We found no evidence of competition between ant- and termite-specialist lizards and dasyurids, thus providing support for our third prediction. Our results revealed evidence of intraguild predation, providing support for our fourth prediction that the mechanism of competition occurs via interference. We explore the consequences of these findings, and possible alternative explanations, in more detail below.

Dasyurid abundance appears to set an upper limit—at the 0.9 quantile—on the diet breadth of nocturnal generalist lizards in our study system. The negative association between dasyurid abundance and the diet breadth of these desert lizards is consistent with our assumption that dasyurids are dominant over nocturnal lizards with generalist diets. As dasyurid abundance increased, nocturnal generalist lizards ate fewer spiders, crickets/grasshoppers and beetles, all of which are largely nocturnal in the desert ecosystem and constitute preferred prey for small dasyurid predators [45,48]. Wolf spiders (family Lycosidae), the most abundant spiders in the study region, are especially favoured by dasyurids, with *S. youngsoni* (the most abundant dasyurid) selecting these while foraging in preference to spiders from other families and in preference to other invertebrates that are potentially available [13,14].

The diets of nocturnal generalist lizards are unlikely to be constrained by factors other than the presence of dasyurid marsupials. In captivity, most invertebrate taxa are acceptable to these lizards [84], and when dasyurid abundance is high, it is reasonable to expect that productivity—and invertebrate abundance, diversity and availability—also should be high. It is further implausible that the diets of nocturnal generalist lizards would appear to be constrained because sample sizes are smaller at high than at low dasyurid abundance; sample sizes were adequate to describe diet breadth irrespective of dasyurid abundance. Another possibility is that the numbers or trappability of the dasyurids fluctuated with season or weather conditions such that their high abundances at times when the diets of the nocturnal generalist lizards were most constrained were more apparent than real. However, this is most unlikely. The pitfall traps that we used are less susceptible to trapping biases than other means of live-trapping [55,69] and thus are likely to reflect actual patterns of abundance. Thus, we conclude that the narrowing of diets found here when dasyurid abundances were high, especially with the taxa that are eaten less being those that are preferred by dasyurids, almost certainly represents competition. It is possible that nocturnal lizards switch the prey that they hunt when dasyurids are abundant, or that heavily depredated invertebrates change their behaviour to reduce their risk of being detected and

eaten (i.e., behavioural resource depression [85]). However, it is perhaps most likely that nocturnal lizards shift the microhabitats where they forage to exploit different invertebrate taxa that occur in different parts of the desert landscape at different times [86,87]. This explanation accords also with theoretical predictions that dominant competitors will cause habitat or microhabitat types—but not prey types—to be dropped from the itinerary of subordinate competitors, with dietary restriction then arising from subordinate individuals gaining access to a limited range of habitat-restricted prey types (i.e., compression hypothesis [88]). Regardless of the mechanism, a shift by nocturnal lizards either in the prey types that they hunt or in the microhabitats where these prey occur would serve to minimise the likelihood of potentially dangerous direct interactions, such as harassment and food-robbing, with dasyurids.

It is unclear whether all species of dasyurids compete with nocturnal dietary generalist lizards. Similar effects on the diet breadth of these lizards were produced for all dasyurids and for the most abundant species, the lesser hairy-footed dunnart (*S. youngsoni*), and it is likely that this abundant species drove the overall results. Three other species of *Sminthopsis* were captured during the study, all larger than *S. youngsoni*, and it is plausible that each would be competitively superior to the nocturnal lizards if their numbers were higher. Three other dasyurid species, all <10 g (Table S3), may have less competitive effect even if their numbers were high, but experimental evaluation is needed to test this speculation. It is unlikely that sexes within the dasyurid species have differential effects on the diets of nocturnal lizards; although slightly more males (56%) than females (44%) were captured over the course of the study, perhaps reflecting longer movements or larger ranges of males [49], females and males are similar in size and in diet [13,45,49]. The largest dasyurid, the brush-tailed mulgara (*D. blythi*), surprisingly, did not affect the diet breadth of nocturnal generalist lizards. This may be due to the relatively low abundance of this species and lower dietary overlap with these lizards compared to other dasyurid species owing to its greater consumption of vertebrate prey [47,89,90]. Alternatively, *D. blythi* suppresses the abundance and activity of the smaller *S. youngsoni* [49,51], and it is therefore possible that higher abundances of *D. blythi* indirectly release nocturnal generalist lizards from competition via their suppressive effect on *S. youngsoni*.

Surprisingly, we found a negative association between the diet breadth of diurnal generalist lizards and the abundance of the ostensibly nocturnal brush-tailed mulgara. Although unexpected, this effect could arise from exploitation competition, which is usually a weaker form of competition than interference competition [91]. The mulgara is the only dasyurid at our study site that digs [92], thus it is possible that mulgaras dig up diurnal invertebrates that burrow under sand or leaf litter at night, thereby reducing their availability for the diurnal lizards. These particular invertebrates would be available for diurnal generalist lizards to eat, but usually not available for nocturnal dietary generalists such as other small dasyurids and geckoes. Another possible explanation for the effect of mulgaras on the dietary breadth of diurnal generalist lizards is that mulgaras may be slightly active by day and restrict lizard diets through interference competition. There is some evidence that mulgaras are active near their burrow entrances and sun bask in the morning, and may also on occasion emerge from their burrows up to 3.5 h before sunset [93,94]. Diurnal activity may also increase when mulgara abundance is high due to intraspecific competition among individuals for prey and the need to increase the time allocated to foraging activity. Although such diurnal activity is likely to be infrequent, it is nonetheless possible that it contributes to the dietary restriction in diurnal generalist lizards.

Alternatively, diurnal lizards may be responding to increased predation risk, as mulgaras occasionally prey upon reptiles including skinks and geckoes [47]. This seems less likely, however, as nocturnal lizards (geckoes), which are at higher risk of potential mulgara predation given their similar activity times, did not alter their diet with increased mulgara abundance. As mulgara abundance increased, diurnal generalist lizards ate more true bugs (Hemiptera), which suggests that these taxa are not the preferred prey of mulgaras or that the lizards shifted to exploiting microhabitats where true bugs were more accessible or abundant.

The diet breadth of ant- and termite-specialist lizards was not affected by changes in overall dasyurid abundance, nor the abundance of the lesser hairy-footed dunnart or mulgara separately. This result was expected as there should be little to no dietary overlap given that dasyurids do not generally eat many ants or termites [45,49]. In addition, while dasyurids may eat termites opportunistically if they encounter them, all species except the very smallest (<9 g) explore foraging paths and microhabitats that maximise their chances of encountering large invertebrates rather than termites and ants [95]. This behaviour would also minimise the likelihood of direct encounters between dasyurids and dietary specialist lizards, even if the latter were active at night. Some species of specialist lizards are also fossorial when active at night [96], and this would reduce their chances of encountering dasyurids still further.

Support for our fourth prediction was adduced by the finding of skink or gecko remains in the diets of three species of dasyurids, whereas no dasyurid remains were found in any lizard diet samples. Although only 17 of 224 dasyurid diet samples contained lizards (7.6%), 12 of these samples contained the remains of geckoes. It was not possible, unfortunately, to determine whether dietary generalist or specialist geckoes had been eaten, but as generalists such as *N. levis* were quite abundant in the study area [84], it is quite likely that some generalists, at least, would have been consumed. As such generalists can be placed in the same generalist insectivore guild as dasyurids, consumption of these lizards by the dasyurids likely represents the most extreme form of interference competition: intraguild killing and predation. It is possible, but very unlikely, that dasyurids consumed the geckoes as carrion because prey movement is an important stimulus in the predation sequence; dasyurids usually eschew carrion [49]. We conclude therefore that nocturnal generalist lizards risk extreme interference competition from dasyurid marsupials while foraging, and respond to this by progressively restricting their diets as dasyurid abundance increases. Further research is needed to identify the cues that nocturnal lizards use to detect increased abundances of dasyurids, and whether lizards focus on prey taxa that dasyurids do not prefer or simply encounter this restricted range of prey by switching to microhabitats or activity times that reduce the risk of encounter with dasyurids.

This study provides valuable evidence of diffuse competition between taxonomically different groups; indeed, our findings appear to be the first to document this interaction between members of the classes Mammalia and Reptilia. While the highly variable and climatically unpredictable desert environment is likely to have a strong influence on the organisation of lizard communities [60,97,98], diffuse competition between insectivorous nocturnal lizards and dasyurids appears to be an important driver of the dietary shifts we observed in lizards. Although we could not measure demographic rates directly here, we assume that diet breadth may serve as a reasonable proxy for demography in that lizards eating restricted and possibly poorer quality diets should have reduced body condition and fecundity. This assumption warrants testing, as does the possibility that lizards in turn have some reciprocal effect on the diets or demography of dasyurids. Understanding the ecological organisation of rich insectivorous reptile and mammal communities is vital to untangling the complex web of interactions between these intraguild species. Furthermore, such understanding would aid effective management and conservation strategies by enabling more accurate predictions of community responses to disturbances that affect particular species or components of the communities. Diffuse competition remains a somewhat forgotten form of competition, but it is, we suggest, a potentially important force that structures many natural communities and merits renewed attention.

In a broader context, it is relevant to consider that the interactions we have described take place within a large and ancient arena that serves as a crucible for wide-ranging ecological interactions: diffuse competition is one of many forces shaping species' assemblages and dynamics in Australia's central deserts. In the first instance, phylogenetic evidence suggests that arid-dwelling marsupials and lizards began to diverge from mesic-adapted ancestors in the early- to mid-Miocene, establishing diverse communities by the end of this epoch [99,100]. This rich desert fauna may have slowed the ingress of rodents, the ancestors of which arrived in Australia around the Miocene-Pliocene boundary 5.3 million years ago [101], allowing lizards and marsupials to continue to radiate in the presence of a rodent fauna that is more depauperate than that of any other desert region [102]. Despite the

rich contemporary marsupial and lizard fauna of arid Australia, it has nonetheless become much less diverse since European arrival: insectivorous and omnivorous bandicoots and two species of dasyurid marsupials have become regionally or completely extinct, and several species of skinks have declined in range [68,103]. Webs of interactions between mammals and reptiles thus are likely to be less complex than they were in the very recent past.

Even with recent losses of species, biotic interactions within communities of ground-dwelling vertebrates in arid Australia are myriad. All species of small dasyurids use burrows constructed by other species, notably agamids and other lizards, and thus are facilitated by components of the same guild with which they compete [51,92]. Larger lizards such as goannas (*Varanus* spp.) in turn depredate dasyurids, often digging them from the shallow burrows that the dasyurids have appropriated from smaller lizards. While there are obviously no benefits to individuals that are killed by goannas, strong selective benefits should accrue to animals that recognise the risk of predation and respond appropriately. Baker and Dickman [49] suggested that dasyurids respond by employing a form of nomadism: individuals range widely in their nightly movements and seldom return to the same burrow, thus avoiding any buildup of odours that could signal their presence to goannas. High mobility, lack of fixed ranges and usually low densities probably reduce the frequency of encounters between species and perhaps facilitate the high local richness of dasyurids in arid regions [104,105]. Within communities of both dasyurids and lizards, trophic and fear-based ecological cascades drive local patterns of species richness, at least at certain times [106]. Conceptual models and structural equation models have been constructed to capture some of these diverse interactions and predict their consequences [51,103]. Our findings here suggest that diffuse competition should be a further and integral component of future models that seek to unravel the complex dynamics of Australia's desert vertebrate communities.

**Supplementary Materials:** The following are available online at http://www.mdpi.com/1424-2818/12/9/355/s1, Table S1: Lizard species list for lizard diet analyses (faecal pellet and stomach samples); Table S2: List of prey taxa identified in lizard faecal pellet and stomach samples; Table S3: List of dasyurid species captured at the study site; Table S4: Results of quantile regression models comparing the 0.5 and 0.9 quantiles of lizard diet breadth (species richness) as a function of dasyurid abundance; Figure S1: Relationships between lizard diet breadth (based on species richness) and dasyurid abundance (all species) at the 0.5 (solid line) and 0.9 (dashed line) quantiles for (a) nocturnal generalist lizards, (b) diurnal generalist lizards and (c) specialist lizards, and (d) mulgara (*Dasycercus blythi*) abundance and diurnal generalist lizards; Figure S2: Non-metric multi-dimensional scaling ordinations of diet composition for nocturnal generalist lizards with (a) total dasyurid abundance, (b) *Sminthopsis youngsoni* abundance, and (c) diurnal generalist lizards with mulgara (*Dasycercus blythi*) abundance.

**Author Contributions:** C.R.D. conceived the idea; all authors designed the methodology, collected and analysed the data and wrote the manuscript. All authors have read and agreed to the published version of the manuscript.

**Funding:** This research was funded by the Australian Research Council, with grant numbers DP0988535 and DP140104621 awarded to C.R.D.

**Acknowledgments:** This study builds on the work of several Honours students, especially Rebecca Drury, Rowena Haynes and Chloe Sato. We thank the members of the Desert Ecology Research Group, especially Bobby Tamayo, and the hundreds of volunteers, students and assistants who assisted with pitfall trapping and scat collection in the field over the years. Thanks to D. and P. Smith, H. and S. Jukes, G. and C. McDonald and Bush Heritage Australia for permission to use the study sites, and Burt Kotler and an anonymous reviewer for insightful comments that improved the final manuscript.

**Conflicts of Interest:** The authors declare no conflict of interest.

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
