# Peer review of "Class Conflict: Diffuse Competition between Mammalian and Reptilian Predators"

_diversity, doi:10.3390/d12090355_

Round 1
Reviewer 1 Report
This manuscript reports on the diets of a Simpson Desert community of insectivores comprised of lizards and small mammals collected over 24 years. Typically, I do not get excited about studies of the diets of vertebrates based on fecal remains or stomach contents (the data of this study come from both). But I make an exception for this one. Among other things, Australia exists to keep ecologists humble. The staggering diversity of reptiles, the depauperate mammal fauna, the environmental variability all make for striking ecosystems. So a data set spanning 24 years with which the authors can correlate diet breadths of lizards with densities of mammalian competitors is deeply compelling. Consequently, I greatly enjoyed reading this manuscript and find it to be a really valuable contribution. My major criticism is that the authors shy away from discussing their study and their results in this larger context. I am sure that other readers would be equally interested in hearing the authors’ thoughts and ideas, and I encourage them to share their speculations with us. Other things that the authors might want to consider are as follows. The authors could take better advantage of foraging theory to separate competitive effects into scramble competition and habitat selection. In my comments on the manuscript and below I discuss this further (spoiler alert: foraging theory suggests that competitive effects are played out mostly through habitat selection and the compression hypothesis). Also, additional insights into competitive interactions might be gained by regressing competitor densities against each other ala Hallet and Pimm (see: Rosenzweig, M.L., Z. Abramsky, B.P. Kotler, and W. Mitchell. 1985. Can interaction coefficients be estimated from census data? Oecologia 66:194-198, for a discussion of some limitations of this approach).
Specific comments:
- Line 10. In similar ways is less important than exploiting similar resources. The key is whether by exploiting the resources they benefit in a manner that increases population growth rate and they make the resources less available to the competitor in a manner that harms the population growth rate of the competitor.
- Lines 16-17. Strongly suggests that lizards and mammals do not rank food items in the same order according to e/h. Otherwise, increased competition would increase diet breadth. Alternatively, habitat selection may have a lot to do with your result, instead, with different types of habitat containing different species of prey. Very interesting! More later.
- Lines 30-35. This goes back all the way to Rosenzweig's 1966 Am. Nat. paper on coexistence in sympatric Carnivora. In this paper, interference and intraguild predation pay prominent roles.
- Lines73-90. What interests me most about this study is that in involves lizards and mammals in Australia. Australia is depauperate in mammals but very species rich in lizards in arid areas. I doubt that this is a coincidence. It is one of the more obvious ways that Australia is so different from other systems in similar climates in other parts of the world. Looking at the interaction of Australian lizards and small mammals is irresistible. This is a prime motivator for your study as far as I am concerned.
- Lines 105-109. What do we expect for small mammals experiencing competition from lizards? Or do you expect it to be one-way? If you do, why? This is an important omission in the manuscript.
- Lines 99-105. This can supplement your predictions. Foraging theory tells us that: a. Diet breadth will be narrower when food is more abundant and broader as food becomes more scarce. Similarly, low population densities correspond to narrow, more selective diets, and populations near K will have broader, more opportunistic diets. Competition will broaden diets for species with shared preferences, especially if they rank prey species similarly. Competition can cause narrowing of diets if competitors have distinct preferences and rank species in opposite order. c. Competition can do the same for habitat selection where having different rankings of habitats is highly likely. That is, competition when there is shared preference will cause habitat use to become broader, while competition when there is distinct preference will cause habitat use to become narrower and more distinct. If this is accompanied by different prey species in different habitat types it will also affect diet breadth and overlap.
- Lines 260-266. Possibly instead of habitat selection, perhaps spiders, crickets, and beetles change their behavior (behavioral resource depression) in response to increasing dasyurid abundances?
- Lines 291-292. Before making that conclusion, wouldn't you want to assess the effect of lizard abundance on dasyurid diet breadth, too?
- Lines 306-307. I think we can pretty much attribute it to changing habitat selection since scramble competition is unlikely to reduce diet breadth. That is the expected outcome from habitat selection so long as habitats differ in the species of prey they hold and lizards and small mammals rank habitats differently in quality (Schoener's compression hypothesis, ARES 1971 I think). It could also be a change in prey behavior that makes them less available to lizard predators (behavioral resource depression, Charnov et al 1976, Amer. Natur.).
- Lines 374-390. As someone whose foundations in ecology were obtained in North America and in the Middle East, I am always delighted by the differences that Australia presents to how I see the world. In that sense, this Discussion is a good start, but I feel there is a lot more to say. Australian deserts are riots of lizard species diversity and abundances. Presumably this has a lot to do with how impoverished the mammalian fauna is. That makes this the results of this study all the more interesting and important. And I sure would welcome a small discussion from the authors on this topic. Continuing on that line, one can view lizards versus mammals in three ways. First, the lack of mammals (including low species diversity) leaves ecological and evolutionary opportunities for lizards to diversify. secondly, the high abundance and diversity of lizards leave little opportunities for small mammals. Thirdly, both insectivorous lizards and insectivorous mammals are abundant and diverse perhaps because arthropods are particularly abundant and diverse perhaps because of Australia's extraordinary variance in ppt and provide ample ecological and evolutionary opportunity for both groups. The interactions within taxa get played out within habitats, and the interactions beween taxa get played out across habitats. You data best fit this last one. In this case, the depauperate nature of the mammal fauna comes across in the rodent fauna (and granivory) , but not the marsupial one. I am sure there are other scenarios that you have thought about. I would love to hear your ideas.
- There are more comments on the manuscript.
Burt P. Kotler
Author Response
Reviewer #1:
This manuscript reports on the diets of a Simpson Desert community of insectivores comprised of lizards and small mammals collected over 24 years. Typically, I do not get excited about studies of the diets of vertebrates based on fecal remains or stomach contents (the data of this study come from both). But I make an exception for this one. Among other things, Australia exists to keep ecologists humble. The staggering diversity of reptiles, the depauperate mammal fauna, the environmental variability all make for striking ecosystems. So a data set spanning 24 years with which the authors can correlate diet breadths of lizards with densities of mammalian competitors is deeply compelling. Consequently, I greatly enjoyed reading this manuscript and find it to be a really valuable contribution. My major criticism is that the authors shy away from discussing their study and their results in this larger context. I am sure that other readers would be equally interested in hearing the authors’ thoughts and ideas, and I encourage them to share their speculations with us. Other things that the authors might want to consider are as follows. The authors could take better advantage of foraging theory to separate competitive effects into scramble competition and habitat selection. In my comments on the manuscript and below I discuss this further (spoiler alert: foraging theory suggests that competitive effects are played out mostly through habitat selection and the compression hypothesis). Also, additional insights into competitive interactions might be gained by regressing competitor densities against each other ala Hallet and Pimm (see: Rosenzweig, M.L., Z. Abramsky, B.P. Kotler, and W. Mitchell. 1985. Can interaction coefficients be estimated from census data? Oecologia 66:194-198, for a discussion of some limitations of this approach).
Response: Thank you for these encouraging comments. In response, we have added some more discussion on foraging theory and how our results conform with it, especially with respect to the key role of habitat in explaining the dietary narrowing that we observed in nocturnal generalist lizards (lines 334-342), as well as two paragraphs outlining where this study fits within the bigger picture of interactions in desert ecosystems (lines 430-461).
With respect to gaining additional insights into competition by regressing competitor densities against each other, we agree this would be an excellent thing to do. Unfortunately, this technique requires robust census data for the putatively competing species, which we do not have. Nonetheless, we note the possibility of using the regression technique and explain the rationale for our different approach in more detail at lines 106-112.
Specific comments:
- Line 10. In similar ways is less important than exploiting similar resources. The key is whether by exploiting the resources they benefit in a manner that increases population growth rate and they make the resources less available to the competitor in a manner that harms the population growth rate of the competitor.
Response: We have deleted 'in similar ways' from the text here.
- Lines 16-17. Strongly suggests that lizards and mammals do not rank food items in the same order according to e/h. Otherwise, increased competition would increase diet breadth. Alternatively, habitat selection may have a lot to do with your result, instead, with different types of habitat containing different species of prey. Very interesting! More later.
Response: Good points, and we completely agree. We have not modified this point in the text of the Abstract (we are at the word limit), but have come back to it in more detail in the Discussion. For example, "it is perhaps most likely that nocturnal lizards shift the microhabitats where they forage to exploit different invertebrate taxa that occur in different parts of the desert landscape at different times" (lines 337-339). And: "This explanation accords also with theoretical predictions that dominant competitors will cause habitat or microhabitat types—but not prey types—to be dropped from the itinerary of subordinate competitors, with dietary restriction then arising from subordinate individuals gaining access to a limited range of habitat-restricted prey types (i.e., compression hypothesis)" (lines 339-342).
- Lines 30-35. This goes back all the way to Rosenzweig's 1966 Am. Nat. paper on coexistence in sympatric Carnivora. In this paper, interference and intraguild predation pay prominent roles.
Response: We have added reference to Rosenzweig's paper.
- Lines73-90. What interests me most about this study is that in involves lizards and mammals in Australia. Australia is depauperate in mammals but very species rich in lizards in arid areas. I doubt that this is a coincidence. It is one of the more obvious ways that Australia is so different from other systems in similar climates in other parts of the world. Looking at the interaction of Australian lizards and small mammals is irresistible. This is a prime motivator for your study as far as I am concerned.
Response: Thank you. We have reworded slightly at lines 86-88, and also added two concluding paragraphs in the Discussion that highlight further some of the oddities of Australian desert ecosystems.
- Lines 105-109. What do we expect for small mammals experiencing competition from lizards? Or do you expect it to be one-way? If you do, why? This is an important omission in the manuscript.
Response: Yes, we did expect the competition to one way, or at least, strongly asymmetrical. The rationale for this is given at lines 95-97 (i.e. dasyurids are much larger and have greater resource demands than lizards), and a validation is provided in the Results. Thus, lizards were often victims of attacks by dasyurids (17 of 224 diet samples), but no dasyurids were found to be eaten by lizards despite an anlysis of 542 diet samples (lines 290-296 and 396-398).
- Lines 99-105. This can supplement your predictions. Foraging theory tells us that: a. Diet breadth will be narrower when food is more abundant and broader as food becomes more scarce. Similarly, low population densities correspond to narrow, more selective diets, and populations near K will have broader, more opportunistic diets. Competition will broaden diets for species with shared preferences, especially if they rank prey species similarly. Competition can cause narrowing of diets if competitors have distinct preferences and rank species in opposite order. c. Competition can do the same for habitat selection where having different rankings of habitats is highly likely. That is, competition when there is shared preference will cause habitat use to become broader, while competition when there is distinct preference will cause habitat use to become narrower and more distinct. If this is accompanied by different prey species in different habitat types it will also affect diet breadth and overlap.
Response: Great points. We have now added more discussion about the relative dietary preferences of the two predator groups, and have discussed the (strong) likelihood that diet narrowing in nocturnal generalist lizards most likely arose because they were competitively pushed into a narrower range of habitats or microhabitats by dasyurids, and in the compressed range of habitats had access to a small range of prey (lines 337-342).
- Lines 260-266. Possibly instead of habitat selection, perhaps spiders, crickets, and beetles change their behavior (behavioral resource depression) in response to increasing dasyurid abundances?
Response: Yes, this is certainly a possibility. We have added this to the Discussion (lines 334-336).
- Lines 291-292. Before making that conclusion, wouldn't you want to assess the effect of lizard abundance on dasyurid diet breadth, too?
Response: It would be fascinating to examine how differences in lizard abundance might affect the diet breadth of dasyurids, but we are not able to carry out this assessment because of our much smaller dasyurid diet sample size and, especially, our inability to gather reliable information on lizard abundance. Thus, we have toned down our conclusion, removing 'supports' so that we say: "The negative association between dasyurid abundance and the diet breadth of these desert lizards is consistent with our assumption that dasyurids are dominant over nocturnal lizards with generalist diets" (lines 312-314).
- Lines 306-307. I think we can pretty much attribute it to changing habitat selection since scramble competition is unlikely to reduce diet breadth. That is the expected outcome from habitat selection so long as habitats differ in the species of prey they hold and lizards and small mammals rank habitats differently in quality (Schoener's compression hypothesis, ARES 1971 I think). It could also be a change in prey behavior that makes them less available to lizard predators (behavioral resource depression, Charnov et al 1976, Amer. Natur.).
Response: We agree, and have added a comment about the importance of habitat shifts under competition in nocturnal generalist lizards, and the reduced access this likely provided in the current study to habitat-restricted prey types, noting that this interpretation accords with the predictions of Schoener's compression hypothesis (lines 339-342).
- Lines 374-390. As someone whose foundations in ecology were obtained in North America and in the Middle East, I am always delighted by the differences that Australia presents to how I see the world. In that sense, this Discussion is a good start, but I feel there is a lot more to say. Australian deserts are riots of lizard species diversity and abundances. Presumably this has a lot to do with how impoverished the mammalian fauna is. That makes this the results of this study all the more interesting and important. And I sure would welcome a small discussion from the authors on this topic. Continuing on that line, one can view lizards versus mammals in three ways. First, the lack of mammals (including low species diversity) leaves ecological and evolutionary opportunities for lizards to diversify. secondly, the high abundance and diversity of lizards leave little opportunities for small mammals. Thirdly, both insectivorous lizards and insectivorous mammals are abundant and diverse perhaps because arthropods are particularly abundant and diverse perhaps because of Australia's extraordinary variance in ppt and provide ample ecological and evolutionary opportunity for both groups. The interactions within taxa get played out within habitats, and the interactions beween taxa get played out across habitats. You data best fit this last one. In this case, the depauperate nature of the mammal fauna comes across in the rodent fauna (and granivory) , but not the marsupial one. I am sure there are other scenarios that you have thought about. I would love to hear your ideas.
Response: Thank you. We have added a couple of paragraphs to address these suggestions, speculating on why insectivorous marsupials and lizards predominate in Australia's deserts rather than rodents, factors that contribute to their extraordinary diversity, and the webs of interactions that extend beyond the marsupials and lizards that we studied in detail in the manuscript. We hope the scenarios that we have sketched are appropriate.
- There are more comments on the manuscript.
Response: Most of the comments made directly on the manuscript have been covered in our responses above, but all others have been addressed also. For example, new references have been added citing work on competition between taxonomically disparate taxa (e.g. larks and gerbils, ants and rodents), we have discussed invertebrates and their distribution in the desert study region, and the possibility that invertebrates shifted their behaviour in response to increased predation risk.
Reviewer 2 Report
Dear Authors:
I think this is a very good paper of interest to readers in a range of scientific disciplines (ecology, herpetology, entomology, and mammalogy). My only concern is with the lack of details concerning the 24-yr sampling regime. It would be beneficial to provide information on the sampling effort during this time period. Was sampling more intense at the beginning rather than middle or near the end of the 24-yr time period? What were the effects of seasons on the abundance of predators and their prey? Something must be said about the sex of the predators. Were females (rather than males) expected to be more abundant in the samples, because they are more involved in foraging for food? During the mating season, males are expected to be more active (than females), therefore appearing more often in pitfall traps?
Do the authors have any information on the propensity of lizards versus mammals to avoid capture in pitfall traps? The rate of capture could be different between the two classes. Would this bias the pitfall data? Are there other techniques, besides pitfall trapping, that could have been used to predict competition or intraguild predation?
Aside: Can you say more about the diversity of arthropods in the pitfall traps? I would expect to see many species of beetles (Coleoptera) in the traps. I think beetles would represent a significant proportion of the items found in the digestive tract and in feces of lizards.
Author Response
Reviewer #2:
I think this is a very good paper of interest to readers in a range of scientific disciplines (ecology, herpetology, entomology, and mammalogy). My only concern is with the lack of details concerning the 24-yr sampling regime. It would be beneficial to provide information on the sampling effort during this time period. Was sampling more intense at the beginning rather than middle or near the end of the 24-yr time period? What were the effects of seasons on the abundance of predators and their prey? Something must be said about the sex of the predators. Were females (rather than males) expected to be more abundant in the samples, because they are more involved in foraging for food? During the mating season, males are expected to be more active (than females), therefore appearing more often in pitfall traps?
Response: We have added more details on the sampling regime to note that there was even sampling intensity over the 24-year study period (line 140) and that sampling effort totalled 187,272 trap-nights (line 240). We also now address the possibility that seasonal or sex effects may have influenced the results (lines 327-332 and 353-356), and conclude that our sampling methods was sufficiently robust that it yielded reliable and consistent results.
Do the authors have any information on the propensity of lizards versus mammals to avoid capture in pitfall traps? The rate of capture could be different between the two classes. Would this bias the pitfall data? Are there other techniques, besides pitfall trapping, that could have been used to predict competition or intraguild predation?
Response: It is quite likely that the trappability of lizards is less than that of dasyurid marsupials simply because lizards have small ranges and may be active for only 3-4 days a week, whereas dasyurids make large nightly movements and have to be active every day. However, when active, the chance that an individual that encounters a pitfall trap will fall into it is probably similar for lizards and mammals. We have added a comment about the reliability of our sampling method (lines 330-332), but note also that any differences in trappability between the two studied taxon groups are actually not very important. We have added new commentary that we could not reliably assess actual lizard abundances due to low rates of capture and recapture, using this to justify our use of a resource factor - food type - rather than any demographic factor - such as abundance, survival, rate of population growth etc - as the key variable to examine for evidence of competition (lines 106-112).
Aside: Can you say more about the diversity of arthropods in the pitfall traps? I would expect to see many species of beetles (Coleoptera) in the traps. I think beetles would represent a significant proportion of the items found in the digestive tract and in feces of lizards.
Response: Thank you, we had overlooked this aspect in the original manuscript. We have now added some comments on the range and types of invertebrates eaten by species in all three of our lizard diet categories (lines 229-238). We have also added two references [86,87] with a comment that different invertebrate taxa most likely occur in different parts (microhabitats) of the desert landscape (lines 337-339).
In addition to these revisions, we have now added details of the several Animal Ethics Committee approvals and Scientific Licence Permits that were obtained over the duration of the study to allow the research to be conducted.